# Evolution of Hydroxytyrosol, Hydroxytyrosol 4-β-d-Glucoside, 3,4-Dihydroxyphenylglycol and Tyrosol in Olive Oil Solid Waste or “Alperujo”

**DOI:** 10.3390/molecules27238380

**Published:** 2022-12-01

**Authors:** África Fernández-Prior, Alejandra Bermúdez-Oria, Juan Fernández-Bolaños, Juan Antonio Espejo-Calvo, Francisco López-Maestro, Guillermo Rodríguez-Gutiérrez

**Affiliations:** 1Instituto de la Grasa, Consejo Superior de Investigaciones Científicas (CSIC), Campus Universitario Pablo de Olavide, Edificio 46, Ctra. de Utrera, km. 1, 41013 Seville, Spain; 2Tecnofood ID Soluciones S.L. C/Aristóteles nº 57, 18.100 Armilla, Granada, Spain; 3San Miguel Arcángel S.A., Paraje La Parrilla, S/N, 23330 Villanueva del Arzobispo, Jaén, Spain

**Keywords:** hydroxytyrosol, 3,4-dihydroxyphenylglycol, tyrosol, antioxidants, olive oil by-products, natural extracts

## Abstract

The main by-product generated from the olive oil two-phase extraction system, or alperujo, is undoubtedly a rich source of bioactive components, among which phenolics are one of the most important. The evolution of four of its main phenolics: hydroxytyrosol (HT), hydroxytyrosol 4-β-d-glucoside (Glu-HT), 3,4-dihydroxyphenylglycol (DHPG) and tyrosol (Ty) was studied over two seasons and in ten oil mills under similar climatological and agronomic conditions, for the first time using organic extraction and high-performance liquid chromatography (HPLC-DAD) determination. The results show that HT (200–1600 mg/kg of fresh alperujo) and Ty (10–570 mg/kg) increase, while DHPG (10–370 mg/kg) decreases only in the last month of the season and Glu-HT (1400–0 mg/kg) decreases drastically from the beginning. This evolution is similar between different seasons, with a high correlation between Glu-HT, HT, and Ty. On the other hand, it has been verified that a mixture of alperujos from all the oil mills, which is what the pomace extractor receives, is a viable source of a liquid fraction which is rich in the phenolics studied through organic extractions and especially after the application of a thermal treatment, obtaining values of 4.2 g/L of HT, 0.36 g/L of DHPG, and 0.49 g/L of Ty in the final concentrated liquid fraction.

## 1. Introduction

The development of the olive oil sector in avoiding the generation of any type of waste has been very positive. The main by-product generated in Spain is the so-called “alperujo”, which is the semi-solid fraction coming from the two-phase olive oil extraction system. The main current use of alperujo is the extraction of pomace oil in pomace extractors, and once the oil has been obtained, the solid is used as biomass for energy production. This by-product has a very high moisture content, around 70%, and its development came as a solution for olive oil mills (OOMs), as it prevented the formation of the liquid fraction from the three-phase extraction system known as “alpechín”. However, the appearance of alperujo posed a problem for the pomace extractors, as they had to dry a solid with a higher degree of humidity and a greater organic load [1]. This problem has forced the olive pomace extraction sector to adopt continuous changes toward promoting better utilization of alperujo [2]. Proof of this can be found in the changes carried out in the last two decades that have allowed the use of dried and defatted alperujo as biomass [3]. Current trends point to the use of thermal treatments to improve the drying of alperujo, saving on drying costs and also allowing a liquid product to be obtained that is rich in bioactive components, mainly phenolics and sugars [4]. These treatments are based on the direct application of steam to reach temperatures of 170 °C during short residence times, and the most widespread is the application of a thermo-malaxation at 60–65 °C for 1–2 h followed by a three-phase extraction system. The most sought-after components include hydroxytyrosol (HT), its derivatives such as hydroxytyrosol 4-β-d-glucoside or oleuropein, and tyrosol [5,6,7]. The main phenolic compounds identified and quantified in alperujo and in the solid obtained in the three-phase extraction system are HT, Ty, and their precursors, mainly oleuropein, which is transformed very rapidly—as early as the malaxation. Therefore, HT and Ty are considered the main simple phenols in these by-products, in quantities of about 300 and 450 ppm, respectively [8]. More recently, however, another phenolic has been discovered that is very similar to HT but with even greater bioactive potential: 3,4-dihydroxyphenylglycol (DHPG), which is also present in considerable amounts [9]. There are many publications showing the biological, antioxidant, anti-inflammatory, anti-modulatory, anti-carcinogenic, anti-platelet, anti-microbial, anti-fungal, etc., activities of these phenolics at different levels, both in vitro and in vivo [10,11,12,13].

Despite the importance of these compounds and all the scientific studies that prove it, their industrial development has been very slow. There is no doubt that the best way to take advantage of such an interesting by-product such as alperujo is through the extraction of these components, followed by the application of bioprocesses or the use of the rest of the organic material in agriculture or animal feed [14,15,16]. Moreover, the extraction of these components makes it possible to reduce the level of toxicity of alperujo, since phenolics are precisely the substances which inhibit micro-organisms, a cause that has prevented the bio-treatment of alperujo for several decades [14]. However, in order to recover these components from the alperujo, it is necessary to study the evolution of these phenolics in the alperujo throughout the season, and even over several seasons, in order to know the optimum treatment times for extracting one phenolic or another, or to find out whether the concentrations at which they are present in the alperujo can justify their recovery. The innovative character of the present work is based on the study, carried out for the first time, to determine the behaviour of the four main phenolics of the alperujo over four points in the season for two consecutive years, as well as to promote obtaining a rich source of these phenols for industrial uses. From a liquid source rich in these phenolics, their purification is simpler. There is an infinity of analytical methods, but few systems are used at an industrial level, mainly based on chromatographic systems, the use of adsorbent materials, solvent extraction, or membranes [6,7,10,17].

Therefore, the aim of the present work was to identify and quantify four of the main phenolics present in alperujo and to study their evolution along four equidistant points in the same season during two consecutive seasons. To do this, ten different mills were sought that shared the same climate and soil type, so that agrological and climatic factors would not make a difference. Finally, the possibility of obtaining a final liquid fraction which could serve as a source of these phenolics was studied.

## 2. Results and Discussion

### 2.1. Moisture Content of Samples

The determination of the moisture content of each of the alperujo samples obtained for the two seasons at four different points within each of them is shown in Table 1. It can be seen that the percentage of moisture ranged around 70% in all cases and that for each olive oil mill (OOM), there was no significant difference throughout each season, as the standard deviation does not exceed 20% in any case, and in two olive mills it was close to 6%. In comparing moisture content rates by date, no significant difference was observed among the OOMs either. It can therefore be said that the way the OOMs operate and the moisture content in the olives do not vary sufficiently to produce significant differences. The oil extraction system used by the 10 olive mills is a continuous two-phase extraction system. Unlike the three-phase system, which is also continuous, there is no need to add water in the horizontal centrifuge or decanter, so the alperujo batches coming directly out of the decanter should be very similar to each other if the olives to be pressed have similar moisture content [18]. However, the alperujo can be altered before reaching the pomace extractor by the addition of other effluents, such as water from the vertical centrifuge that cleans the oil after the decanter, or even other types of water generated in the oil tanks prior to storing the final olive oil [19]. These types of additions increase the degree of moisture, and can increase it to values close to 80%, although this has not been observed in any of the olive mills. This is due to the fact that they all work in a very similar way and that the starting olives must also have a very similar degree of moisture content, since they are all close to each other.

### 2.2. Phenolic Profile

Figure 1 shows two chromatograms obtained by HPLC-DAD at 280 nm. (A) is a typical phenolic profile at the beginning of the season, while (B) is from the end of the season, where it is observed that the HT content is higher while GLu-HT disappears.

### 2.3. Evolution of the Main Simple Phenolic Compounds

The evolution of the most active and important simple phenolics in the alperujo was studied throughout the two seasons and at four equidistant points within each of them. 

Figure 2 shows the behaviour of hydroxytyrosol and its glycoside. A clear evolution can be observed: for HT, the concentration increases during the first three months of the season and in the fourth month there are mills where the concentration continues to increase or even decreases, although the tendency is to stabilize. The behaviour of hydroxytyrosol 4-β-d-glucoside is different. It shows its highest value for all the samples at the beginning of the season and drops significantly in the second month, after which it either stabilizes or continues to decrease.

In contrast to reports by other authors [20] where the concentration of Glu-HT increased in olive pulp with increasing maturity, in this study it was observed that in the case of alperujo it decreased. This is due to the fact that in the pulp the secoiridoid phenolics are transformed into simpler phenolics such as Glu-HT, and that this is more easily hydrolysed during the milling and malaxation phases, generating a glucose and HT [21]. In the less-mature samples, which are those processed at the beginning of the season, the most-abundant phenolics are secoiridoids such as oleuropein or ligustroside, which in the milling and malaxation phases have more affinity to being transformed enzymatically into their derivatives such as oleacein and oleocanthal, among others [22].

As has already been observed, the data between seasons are very different, although the evolution was similar. Figure 3 shows whether the differences among the different olive mills for the same year were significant or not in the case of HT and Glu-HT. For HT, there were two olive mills with values between 400 and 500 mg/kg, significantly different from the rest with much higher values. The highest range in HT concentrations was about 700–800 mg/kg, or about 0.08% HT in fresh alperujo at an average moisture content of 70%. In the case of Glu-HT, although there were apparently two with much lower average values, no significant difference was observed. This is due to the great variability in the concentrations found for these olive mills. As can be seen in Figure 2, this variability was due to the large difference in concentration at the beginning of the season compared to the rest of the season since, as indicated above, Glu-HT is easily hydrolysed, helping to increase the concentration of free HT.

The data obtained on the evolution of DHPG between the four points obtained for the different seasons are shown in Figure 4. The differences are remarkable between the values for the first season, where most of the values are below 100 mg/kg of fresh alperujo, and the second season, where all values are between 100 and 350 mg/kg. This is the first time that DHPG values have been reported between seasons. It is also curious that in the case of Ty, its behaviour is different, since in the first season the range of concentrations found was much wider, reaching values exceeding 500 mg/kg; while in the second season most of these values were below 200 mg/kg. The precursors of the two phenolics are distinct. While that of tyrosol is primarily ligustroside, those of DHPG have been identified as verbascoside and isoverbascoside (β-hydroxyacteoside and β-hydroxyisoacteoside) and 2″-hydroxyoleuropein [23]. In contrast to the other phenolics, DHPG has recently been described as a phyto-regulatory substance [24]. That is to say, it has the property of enhancing or decreasing plant growth. This is the behaviour of the so-called allelopathic substances, which are synthesized by the plant to improve its growth and that under certain circumstances can be increased in order to solve growth problems or to prevent the growth of other competing species. Changes in climatic conditions are known to alter the formation of phenolics, which act primarily as a defence against water or light stress and against pests or infections [25]. In the case of DHPG, it may be that there are other factors that cause the synthesis of its precursors to increase, hence the differences observed between the two seasons. In the case of Ty, the variation between seasons is also quite wide, unlike HT or its glycoside. In general, the differences in phenolic content for the same cultivar depend on agronomic variations and the incidence of light and water in each year. In drier years, there is a stress that causes the phenolic content to increase, as well as if the plant has been attacked by phytopathogens or even insects. However, in this case the difference found with DHPG or Ty is more important, which could be due to the similarity of DHPG with phytohormones and to the fact that Ty precursors are present in smaller quantities in olives and can undergo these changes more significantly [9,10,12,23].

In the case of comparing the results obtained from the ten olive mills in the same year (Figure 5), it can be seen that for DHPG there are two olive mills that have higher average values, but only four of them are significantly different from the others. It can be said that there are not many differences among them, and quite high mean values were obtained for the second season. In the case of Ty, there are more significant differences, with three different ranges: one of 50–60 mg/kg, another of 90–110 mg/kg, and another with average values. 

### 2.4. Correlation Values between Mean Values 

The similarities among the values obtained at different times within the same season and between the two seasons were studied. To do this, the correlation values among the averages of all the OOMs were determined, which would represent an approximation of what the pomace extractor receives, assuming that all the OOMs contributed the same amount of alperujo.

Table 2 shows the correlation coefficients among the four phenolic compounds in the first and second season, as well as the correlation coefficients between the two seasons for each phenolic. In the first season, a high positive correlation between HT and Ty concentration was observed, as both increased similarly over time. There was a high negative correlation between the concentration of Glu-HT and Ty, which does not indicate that the latter was transformed into Ty, but rather that while the former increased, the latter decreased in a correlated manner. In the second season, the same was observed, to which a high negative correlation between Glu-HT and HT must be added, which is more logical since Glu-HT is one of the precursors of HT. By comparing the values for each phenolic between the two seasons, a high correlation for all phenolics is shown, except for HT. This may be due to the fact that at the end of the first season there was a decrease in HT that did not occur in the second season, which may have been caused by some kind of thermal degradation due to an increase in temperature or oxidative conditions in the first season as opposed to the second. Apart from this difference, and the fact that the values are different from one season to another, it can be confirmed that the behaviour is very similar: Glu-HT decreases while HT and Ty increase over time, and DHPG was maintained in the first three months and diminished in the last one.

Many studies have been carried out on the evolution of the main phenolics in wastewater from the three-phase extraction system and have shown how HT and Ty increase with time [26]. However, there are hardly any studies showing such an evolution in the alperujo, where HT and Ty have also been shown to increase [27]. In the case of Glu-HT, it has only been shown to increase with fruit ripening. Therefore, this study is the first to show the evolution of Glu-HT and DHPG throughout the season and the importance of the moment at which the alperujo must be treated in order to obtain a higher concentration of these phenolics.

### 2.5. Liquid Source of Phenolic Compounds

The potential of alperujo as a source of phenolic compounds with high biological activity has been studied by means of a test of extraction of these compounds to the liquid phase, from which their subsequent purification is possible. An initial, direct extraction to obtain the vegetation water was carried out. This extraction had problems associated with the difficult separation of the phases. The centrifugation phase was repeated several times and the liquid phase took more than 24 h to be filtered, changing the filter paper several times. The second extraction was carried out with organic solvent, and although it did not take as long to centrifuge and filter, the phase separation was also slow. The third extraction was carried out using one of the heat treatment systems currently in operation in pomace extractors to improve the utilisation of the alperujo [4]. The thermal treatment was the only system in which the separation of the liquid phase was effective, resulting in a clarified liquid phase in a single centrifugation and without the need for filtration. The concentration data for the phenolics studied are shown in Figure 6. The separation of the vegetation water gave the lowest concentrations of the four phenolic compounds. This fraction is the most similar to the wastewater obtained from the three-phase oil extraction system, also called alpechín. The concentration is close to that reported by other authors for three-phase alpechín [28]. Despite reaching a concentration of 150 ppm HT, the concentration of phenolics was very low, which, together with the difficulty in filtration, would make their removal very difficult. The use of vegetation water would be possible by using the alpechín or olive oil wastewater generated from the three-phase extraction process, and only by applying subsequent phenolic extraction and concentration systems such as solvent extraction, membranes, or chromatographic adsorption systems, all of which involve high investment and operating costs [6].

The phenolic values obtained from the methanol:water mixture were very similar to the analytical values presented previously. This is due to the fact that the same type of solvent was used. Solvent extraction has many environmental drawbacks. In the case of methanol, it cannot be used for food, so ethanol, which is widely used for natural extracts, would have to be used.

In the case of the application of thermal treatment, the values are much higher. This is due to the fact that the phenolics bind to the cell wall material during the milling and malaxation process, which makes their extraction difficult and makes it necessary to apply severe treatments in order to solubilise them. At the same time, these treatments not only solubilise but also hydrolyse the glycosylated phenolics to their simple phenolics, as is the case for HT, Ty and DHPG. In the case of Glu-HT, it disappears after heat treatment as it is one of the HT precursors that are hydrolysed at that temperature. DHPG seems to be affected by heat treatment at 170 °C, which lowers its concentration, as reported in previous work [29].

The data obtained for the three extractions clearly show that alperujo is an important source of phenolics with significant bioactivity, but for its recovery it is necessary to apply systems with sufficient severity to promote the solubilisation of these phenolics, the hydrolysis of the more complex phenolics, and to improve the separation of the phases. This not only allows for obtaining a liquid source that is rich in phenolics, but also a solid with a lower degree of moisture content and lower phenolic load, and therefore more easily exploitable [4].

## 3. Materials and Methods

### 3.1. Materials

The characterisation studies focused on alperujo samples treated in the facilities of San Miguel Arcángel, S.A., located in Villanueva del Arzobispo, Jaén, Spain. San Miguel Arcángel is the biggest pomace extractor plant in the world. The samples came from ten olive oil mill cooperatives in four different periods of the season for two consecutive seasons (2012–2013 and 2013–2014). The season begins in November and usually lasts four or five months, depending on the year. The samples were therefore taken every month starting in the middle of November. The main variety used was picual, followed by a small amount of arbequina. The ten cooperatives from which the alperujo samples were taken are all within the province of Jaén and within a radius of less than 100 km, so the atmospheric and agronomic conditions are not very different from one another. 

### 3.2. Chemicals

3,4-dihydroxyphenylglycol was obtained from Sigma-Aldrich (Deisenhofer, Germany). Tyrosol was obtained from Fluka (Buchs, Switzerland) and hydroxytyrosol was obtained from Extrasynthese (Lyon Nord, Geney, France). Hydroxytyrosol 4-β-d-glucoside and trifluoroacetic acid (TFA) were purchased from Sigma-Aldrich (Madrid, Spain). Acetonitrile (HPLC grade) was obtained from Panreac Quimica S.A. (Barcelona, Spain). Ultrapure water was obtained from a Milli-Q water system (Millipore, Milford, MA, USA).

### 3.3. Determination of Moisture Content

To determine the moisture content in the alperujo, approximately 250 mg were weighed in a crucible, and the sample was placed in an oven at 105 °C for 8 h, after which the crucible was left to cool in a desiccator until constant weight was reached.

### 3.4. Analytical Extraction of Phenolics

The phenolic compounds present in the alperujo samples were extracted following the method described by Obeid et al. [30]. The alperujo was put into contact with 80% methanol in water under gentle agitation at room temperature for 30 min, using 15 mL of methanol for every 10 g of sample. The extraction was repeated twice, using 10 mL of 80% methanol for every 10 g of sample. The fractions were filtered through a Buchner filter paper, pooled, and the methanol evaporated to obtain an aqueous extract.

### 3.5. Obtention of the Phenolic-Rich Liquid Phase

Samples of alperujo (0.6 kg) from each of the ten mills were mixed to simulate a typical mixture usually treated by the pomace extractor. An amount of 6 kg was divided into three 2 kg batches. The first batch was centrifuged at 4000× *g* (Sigma 2–16K, Osterode, Germany) for 5 min, and paper-filtered under vacuum. The second batch was treated with 2 L of a methanol:water mixture (4:1 (*v*/*v*) and subsequently centrifuged under the same conditions as the previous batch. The last batch was subjected to a heat treatment at 170 °C in a 100 L capacity reactor with indirect steam supply for one hour to facilitate greater solubilisation of phenolics on the one hand, and on the other, to improve phase separation. Once treated, it was centrifuged in the same manner as the two previous batches. All extracts were brought to dryness under vacuum, dissolved in 0.5 L of water, and finally stored at 4 °C for further extraction.

### 3.6. Analysis by High-Performance Liquid Chromatography

In order to identify the main single phenolic compounds in the alperujo samples, a high-performance liquid chromatography (HPLC) system (Hewlett-Packard 1100 series equipped with a diode array detector (the wavelength used for quantification was 280 nm) and an Agilent 1100 series automatic injector, which allows for 20 µL of sample) was used. The chromatographic column employed was a Teknokroma Tracer Extrasil OSD2 with a particle size of 5 µm, an internal diameter of 250 mm, and length of 4.6 mm. The mobile phases were 0.01% trichloroacetic acid in water and pure acetonitrile. The gradient for a total run time of 55 min was: 95% A initially, 75% A at 30 min, 50% A at 45 min, 0% A at 47 min, 75% A at 50 min, 95% A at 52 min until the run was completed. The quantification and identification were made using commercial standards of all the phenolics studied by comparing the retention times with those of reference compounds and recording the UV spectra in the range of 200–360 nm. A standard solution was prepared for each phenol at a concentration of 1000 mg/L by dilution of the standard in distilled water. The quantification was performed by means of a five-point regression curve in triplicate of individual stock solutions of each phenolic in the range of 10 to 1000 mg/L. The results were expressed as mg of each phenolic per g of fresh alperujo for the alperujo samples, or per litre of liquid phase for the obtained liquid source of phenolics. 

### 3.7. Statistical Analysis of the Results Obtained

In order to establish statistical differences among the main single phenolics analysed, a multivariate comparison was carried out using the Statgraphics Plus program Version 2.1. A multivariate analysis of variance (ANOVA) followed by Duncan’s multiple comparison test was carried out for the different olive oil mills. The significance level was set at *p* < 0.05.

## 4. Conclusions

Throughout two seasons, the evolution of four of the most important phenolics from the alperujo of ten different olive oil mills taken at four points in each season was studied. This study shows the similarity in the evolution of the phenolics studied between different seasons, although with different values. It also shows for the first time the evolution of Glu-HT and especially of DHPG as one of the most promising phenolics due to its bioactive potential, highlighting the importance of not waiting until the end of the season for its recovery. Depending on the type of phenolics sought, one time of the season is more interesting than another. Thus, if the aim is to obtain DHPG, it is best to extract it in the second and third month of the season, just in the middle, while if HT or Ty is sought, it is best to extract them in the final stages of the season. In the case of Glu-HT, as it is hydrolysed so quickly, it should either be obtained at the beginning of the season or HT should be sought as the product of its hydrolysis.

The evolution of the phenolics shows the dates when the alperujo is an ideal source of recovery for each of them. It is also important to point out that one of the greatest difficulties that this by-product presents is the necessary application of a pre-treatment, as has already been extensively studied in the literature. These pre-treatments aim to achieve the solubilization of the research components at the same time as making phase separation possible at an industrial level. In this sense, heat treatment would be a good option for the recovery of the most active phenolics such as HT, DHPG, and Ty.

Further studies to increase the content of these phenolic compounds of great functional interest will be necessary. Thus, in order to improve the processes that aim to improve the quality of olive oil, they must also improve the quality of the by-products, such as olive pomace, by increasing and facilitating the extraction of their bioactive compounds. It is very important to emphasise that this type of study is only the beginning, since in order to make the extraction and use of these components possible, the synergy of different technologies must be found in order to achieve the integral use of this by-product [7,14,31].

## Figures and Tables

**Figure 1 molecules-27-08380-f001:**
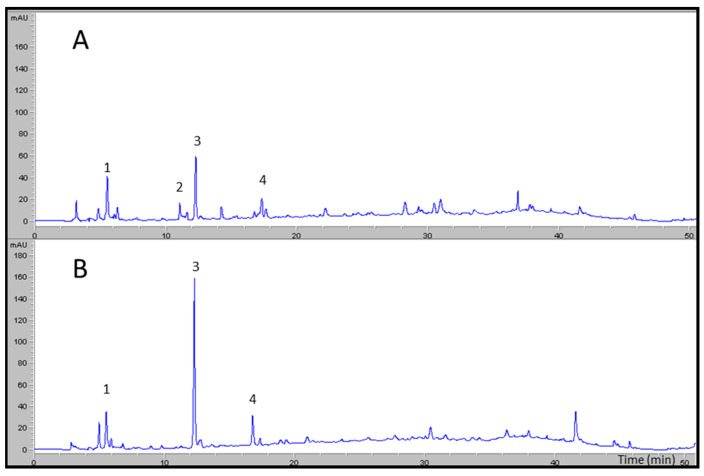
Profiles obtained by HPLC-DAD at 280 nm for two samples: (**A**) alperujo obtained at the beginning of the first season and (**B**) alperujo obtained at the middle of the second season, identifying 3,4-dihydroxyphenylglycol (1), hydroxytyrosol 4-β-d-glucoside (2), hydroxytyrosol (3), and tyrosol (4).

**Figure 2 molecules-27-08380-f002:**
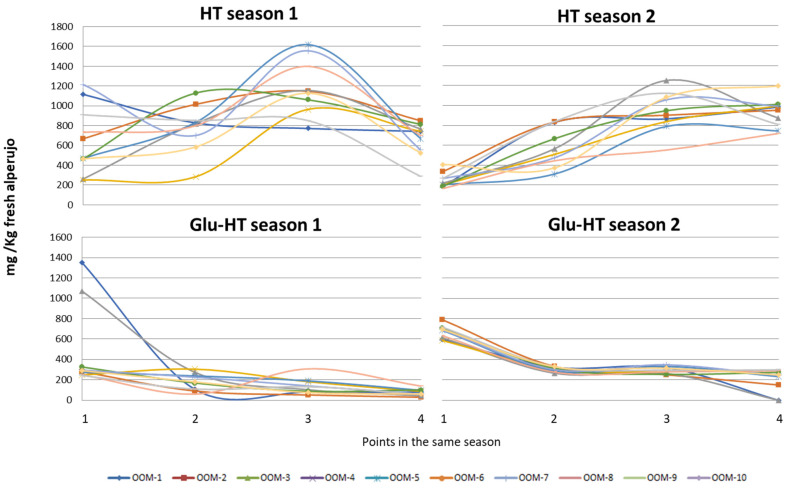
Evolution of the concentration of hydroxytyrosol (HT) and hydroxytyrosol 4-β-d-glucoside (Glu-HT) in the alperujo from the ten olive mills at four points of the season for two consecutive years.

**Figure 3 molecules-27-08380-f003:**
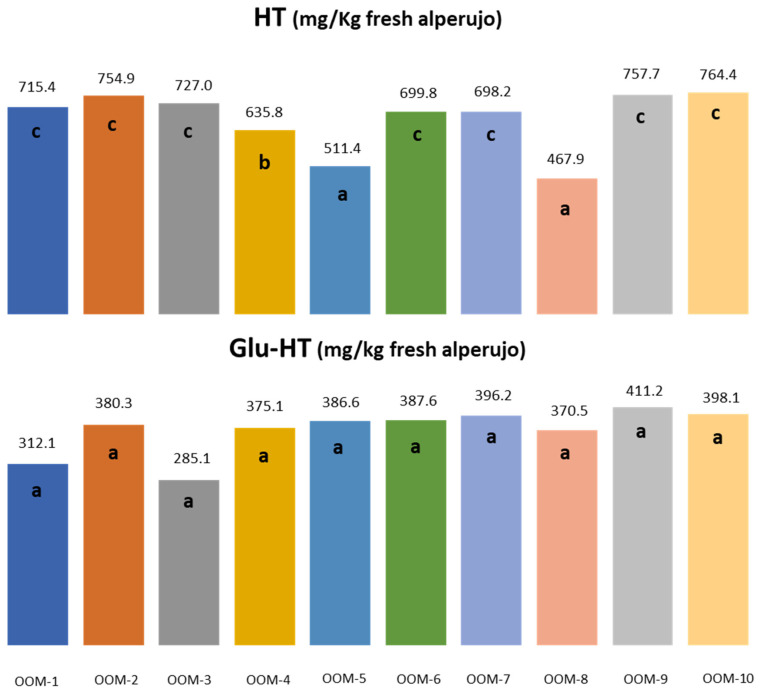
Average concentration of hydroxytyrosol (HT) and hydroxytyrosol 4-β-d-glucoside (Glu-HT) obtained in the second campaign for the ten mills. Means with the same letters were not significantly different, *p* < 0.05.

**Figure 4 molecules-27-08380-f004:**
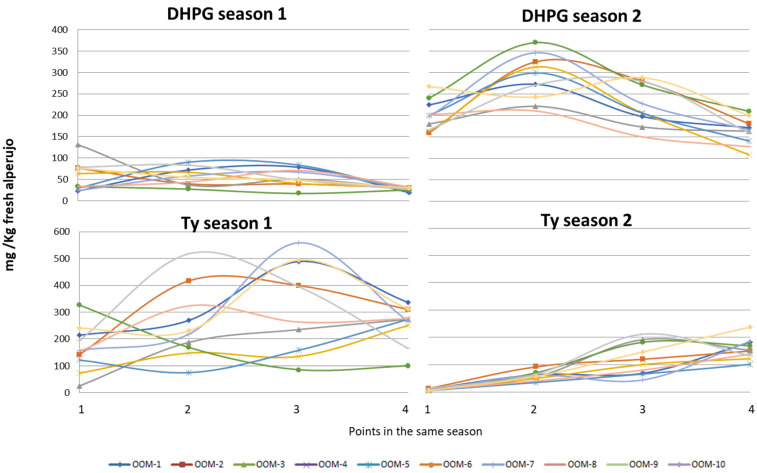
Evolution of the concentration of 3,4-dihydroxyphenylglycol (DHPG) and tyrosol (Ty) in the alperujo of the ten olive mills at four points of the season for two consecutive years.

**Figure 5 molecules-27-08380-f005:**
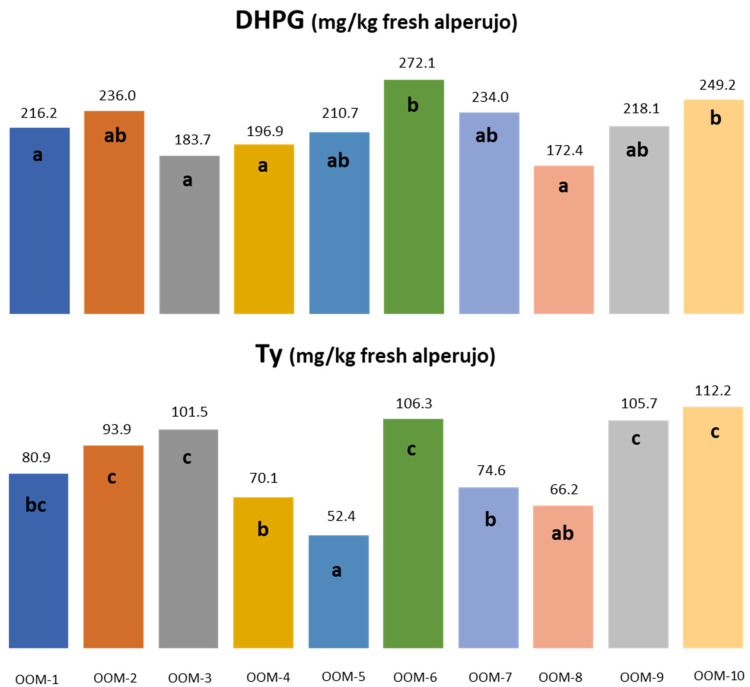
Average concentration of 3,4-dihydroxyphenylglycol (DHPG) and tyrosol (Ty) obtained in the second season for the ten mills. Means with the same letters were not significantly different, *p* < 0.05.

**Figure 6 molecules-27-08380-f006:**
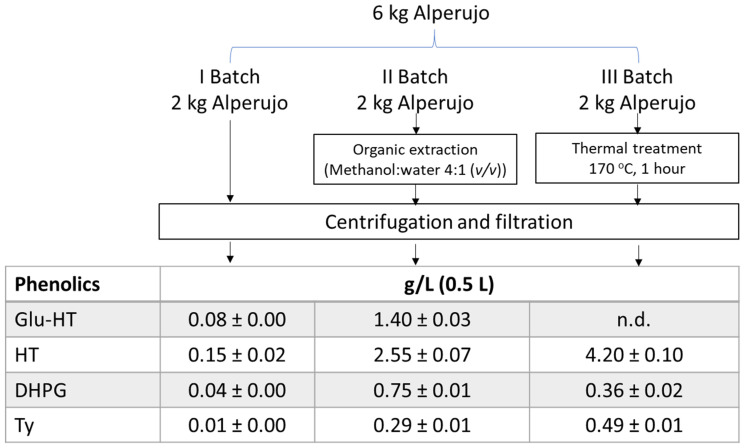
Extraction of liquid phenolic source by direct, organic, and thermal process. Concentration of phenolics in each liquid source obtained. Hydroxytyrosol (HT), hydroxytyrosol 4-β-d-glucoside (Glu-HT), 3,4-dihydroxyphenylglycol (DHPG) and tyrosol (Ty). Not detected: n.d.

**Table 1 molecules-27-08380-t001:** Moisture content (%) of the alperujo samples obtained from ten olive oil mills (OOMs) at four points in the season for two different and consecutive campaigns, showing the average values and standard deviation (SD).

OOM	Season 1 (% Moisture)						Season 2 (% Moisture)					
	1	2	3	4	Average	SD	1	2	3	4	Average	SD
OOM1	67.7	68.7	69.9	70.4	69.2	1.2	67.7	68.7	69.9	70.4	69.2	1.2
OOM2	69.4	72.0	74.1	73.1	72.2	2.0	69.4	72.0	74.1	73.1	72.2	2.0
OOM3	72.4	74.8	73.0	75.2	73.9	1.4	72.4	74.8	73.0	75.2	73.9	1.4
OOM4	68.6	69.2	67.5	68.9	68.6	0.7	68.6	69.2	67.5	68.9	68.6	0.7
OOM5	68.7	70.4	67.0	71.0	69.3	1.8	68.7	70.4	67.0	71.0	69.3	1.8
OOM6	69.9	62.9	69.4	72.3	68.6	4.0	69.9	62.9	69.4	72.3	68.6	4.0
OOM7	68.1	65.8	67.8	69.9	67.9	1.7	68.1	65.8	67.8	69.9	67.9	1.7
OOM8	68.8	64.8	68.5	74.6	69.2	4.0	68.8	64.8	68.5	74.6	69.2	4.0
OOM9	67.1	69.8	68.3	67.1	68.1	1.3	67.1	69.8	68.3	67.1	68.1	1.3
OOM10	70.4	71.6	69.6	72.5	71.0	1.3	70.4	71.6	69.6	72.5	71.0	1.3
Average	69.1	69.0	69.5	71.5			69.1	69.0	69.5	71.5		
SD	1.5	3.6	2.3	2.5			1.5	3.6	2.3	2.5		

**Table 2 molecules-27-08380-t002:** Correlation coefficient values among the mean values of the ten OOMs for the values of each phenolic in each of the seasons and the correlation coefficient values for each phenolic by comparing the two seasons. Hydroxytyrosol (HT), hydroxytyrosol 4-β-d-glucoside (Glu-HT), 3,4-dihydroxyphenylglycol (DHPG), and tyrosol (Ty).

**Season 1**	**DHPG**	**Glu-HT**	**HT**	**Ty**
DHPG	1.00	0.57	0.35	−0.24
Glu-HT	0.57	1.00	−0.37	**−0.89**
HT	0.35	−0.37	1.00	**0.75**
Ty	−0.24	**−0.89**	**0.75**	1.00

**Season 2**	**DHPG**	**Glu-HT**	**HT**	**Ty**
DHPG	1.00	−0.05	−0.19	−0.39
Glu-HT	−0.05	1.00	**−0.91**	**−0.88**
HT	−0.19	**−0.91**	1.00	**0.97**
Ty	−0.39	**−0.88**	**0.97**	1.00

	DHPG	Glu-HT	HT	Ty
Seasons	**0.74**	**1.00**	0.53	**0.81**

## Data Availability

The data are available from the corresponding author upon reasonable request.

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
