# Peer review of "Evolution of Hydroxytyrosol, Hydroxytyrosol 4-β-d-Glucoside, 3,4-Dihydroxyphenylglycol and Tyrosol in Olive Oil Solid Waste or “Alperujo”"

_molecules, 2022, doi:10.3390/molecules27238380_

Round 1
Reviewer 1 Report
The article is novel and informative for readers. It will be interesting to academicians and the industry. Therefore, I recommended publishing after minor revisions.
I have the following suggestions to improve the quality of the article.
Comment #1:
The abstract is long. Please, Modify and add the main results….
Comment #2: Introduction
Please add some articles for explaining the other methods like SFE...
Comment #3:
The keywords should be modified ...
Comment #5: Introduction
Please add a table for abbreviations …
Comment #9:
The paper should edit by a native speaker.
Comment #11:
The authors should rewrite the conclusion….
Author Response
Reviewer 1
The article is novel and informative for readers. It will be interesting to academicians and the industry. Therefore, I recommended publishing after minor revisions.
Response: I have the following suggestions to improve the quality of the article.
Comment #1:
The abstract is long. Please, Modify and add the main results….
Response: The abstract has been shortened and the range of values of the four main phenols has been inserted as one of the main results.
Comment #2: Introduction
Please add some articles for explaining the other methods like SFE...
Response: The following paragraph has been inserted in the text using four references: From the liquid source enriched in these phenols, their purification is simpler. There is an infinity of analytical methods, but few systems used at industrial level, mainly based on chromatographic systems, the use of adsorbent materials, solvent extraction, or membranes [6,7,10,17].
Comment #3:
The keywords should be modified ...
Response: It has been modified by: Hydroxytyrosol, 3,4-dihydroxyphenylglycol, tyrosol, antioxidants, olive oil by-products, natural extracts
Comment #5: Introduction
Please add a table for abbreviations …
Response: All the abbreviations have been described at the beginning, the four abbreviations for the four phenols (HT, Glu-HT, Ty and DHPG), liquid chromatography (HPLC-DAD), olive oil mills (OOM) and for the tables the standard deviation (SD) and the non-detected (nd) appear. From our point of view, they are easy to follow without the need for a table.
Comment #9:
The paper should edit by a native speaker.
Response: The entire text of the manuscript has been edited by a native speaker.
Comment #11:
The authors should rewrite the conclusion….
Response: The conclusions have been improved and rewritten.
Reviewer 2 Report
In the present work, the relative concentration of phenolic compounds contained in olive oil solid wastes (alperujo) sampled at 4 different times during two consecutive seasons was measured. The results show that the relative concentration of phenolic compounds follows a similar behavior in both seasons, although with small variations. This temporal variation in the concentration of phenolics throughout a season offers the possibility of choosing the best harvesting period depending on the desired phenolic compound. The present research provides relevant information in terms of the utilization of solid wastes from olive oil extraction as a source of molecules of biological or industrial interest. In this sense, I judge that it can be considered for publication in "Molecules"; however, it omits valuable information on the reproducibility of the method of analysis, as well as suffering from several inaccuracies. Therefore, the following recommendations should be considered before its possible publication in Molecules:
1. Line 77: Replace “od” by “of”
2. Line 135: HT can not be hydrolyzed, it must be Glu-HT.
3. Lines 125-136: This paragraph is repeated in lines 141-152. I believe it is.
4. Figure 2: Meaning of a,b,c letters inside bar charts? The legend “Different signs imply significant differences” lacks precision, standard-deviation error bars can be considered instead. A similar action is recommended for Figure 4.
5. Line 165. Letter “b” in b-hydroxyacteoside should be changed to “beta” or its corresponding Greek letter. Do the same for b-hydroxyisoacteoside.
6. Figure 5: Meaning of n.d.?
7. Line 249: Change “phenol values” to “phenolic values”
8. Line 258: Change “the phenols” to “the glycosylated phenols”
9. Line 260: Replace “hydrolyses” by “hydrolysed” or “hydrolized”
10. Line 308: Replace “once” by “Once”
11. Lines 317-318: The internal diameter and length of the column must be clearly specified in “internal diameter of 25 x 0,46”.
12. Line 318: Replace “mill-Q” by “milli-Q”
13. Line 318: Acetonitrile and water ratio?
14. Section 3.6 should specify whether the separation was carried out under isocratic or gradient conditions, the retention times of the standards or of the phenolic compounds of interest present in "alperujo". The solvent flow rate, detector type (if UV, provide wavenumber), elution temperature, among others, should also be indicated.
15. It is strongly recommended that the authors provide the most representative chromatograms, either in an additional figure or as supplementary material.
Author Response
Reviewer 2
In the present work, the relative concentration of phenolic compounds contained in olive oil solid wastes (alperujo) sampled at 4 different times during two consecutive seasons was measured. The results show that the relative concentration of phenolic compounds follows a similar behavior in both seasons, although with small variations. This temporal variation in the concentration of phenolics throughout a season offers the possibility of choosing the best harvesting period depending on the desired phenolic compound. The present research provides relevant information in terms of the utilization of solid wastes from olive oil extraction as a source of molecules of biological or industrial interest. In this sense, I judge that it can be considered for publication in "Molecules"; however, it omits valuable information on the reproducibility of the method of analysis, as well as suffering from several inaccuracies. Therefore, the following recommendations should be considered before its possible publication in Molecules:
- Line 77: Replace “od” by “of”
Response: It has been replaced.
- Line 135: HT cannot be hydrolyzed, it must be Glu-HT.
Response: It has been corrected.
- Lines 125-136: This paragraph is repeated in lines 141-152. I believe it is.
Response: That is true, the repeated paragraph has been removed.
- Figure 2: Meaning of a,b,c letters inside bar charts? The legend “Different signs imply significant differences” lacks precision, standard-deviation error bars can be considered instead. A similar action is recommended for Figure 4.
Response: The legend has been changed to explain it better: “Means with the same letters were not significantly different, p < 0.05”. In Material and method, the following sentence explain how it was made and its precision: “Duncan's multiple comparison test was carried out for the different olive oil mills. The significance level was set at P < 0.05”.
- Line 165. Letter “b” in b-hydroxyacteoside should be changed to “beta” or its corresponding Greek letter. Do the same for b-hydroxyisoacteoside.
Response: It has been changed. Thanks
- Figure 5: Meaning of n.d.?
Response: It has been explained in figure food.
- Line 249: Change “phenol values” to “phenolic values”
Response: The word "phenol" has been changed throughout the manuscript to "phenolic" is more correct, as the reviewer rightly points out.
- Line 258: Change “the phenols” to “the glycosylated phenols”
Response: It has been changed.
- Line 260: Replace “hydrolyses” by “hydrolysed” or “hydrolized”
Response: It has been corrected.
- Line 308: Replace “once” by “Once”
Response: It has been also corrected.
- Lines 317-318: The internal diameter and length of the column must be clearly specified in “internal diameter of 25 x 0,46”.
Response: It has been changed to: “internal diameter of 250 mm and length of 4.6 mm”.
- Line 318: Replace “mill-Q” by “milli-Q”
Response: It has been replaced.
- Line 318: Acetonitrile and water ratio?
Response: It has been specified in the text.
- Section 3.6 should specify whether the separation was carried out under isocratic or gradient conditions, the retention times of the standards or of the phenolic compounds of interest present in "alperujo". The solvent flow rate, detector type (if UV, provide wavenumber), elution temperature, among others, should also be indicated.
Response: It has been also specified in this section.
- It is strongly recommended that the authors provide the most representative chromatograms, either in an additional figure or as supplementary material.
Response: A new figure (Figure 1) with two representative chromatograms have been insert in the text.
Reviewer 3 Report
-
Manuscript ID molecules-2017369
“Evolution of hydroxytyrosol, hydroxytyrosol 4-β-D-glucoside, 3,4-dihydroxyphenylglycol and tyrosol in olive oil solid waste or “alperujo”
The subject of the present manuscript is interesting and presents scientific relevance. However, I missed innovation in the present study, what is innovative about your work compared to other works conducted in this line of research. It is important to emphasize the innovative character of your work. In addition, several aspects must be taken into account during the writing of the text.
Below I indicate some points of improvement / suggestions.
- Line 81: Olive oil mill (OOM)
- The authors should detail in the introduction, which are the studies already published in the literature on phenolic composition in the matrix in question (olive pomace)
- Authors are encouraged to detail the methodology, explaining how the quantification of phenolic compounds was performed. Are they expressed in calibration curves, expressed in mg tyrosol equivalent as recommended by the IOC?
- The behavior observed (season 1 and 2) in the concentration of the phenolic compounds in question should be further discussed.
- Figure 1: mg T/Kg fresh alperujo? The graphs are confused. It is difficult to understand the results. Please, think in the way to clarify better the results obtained.
- All graphs and tables deserve improvement. The results obtained may even be interesting, but both the methodologies and the results obtained are lacking in detail.
- Sometimes mention “olive oil solid waste”, “pomace”, “alperujo”, please standardize the writing.
- 301: It is not good to start sentences with numbers.
- Figure 4: The results are expressed in mg/L or mg/kg?
- How were the quantifications of phenolic compounds performed?
- References must be formatted according to the journal's rules
- All abbreviations must be cited in the text
- All figures and tables must be cited in the text
- It is missed the critical point of view in conclusion section.
- The cited references are relevant, but I miss a critical look at the results indicated by the literature.
Author Response
Reviewer 3
The subject of the present manuscript is interesting and presents scientific relevance. However, I missed innovation in the present study, what is innovative about your work compared to other works conducted in this line of research. It is important to emphasize the innovative character of your work. In addition, several aspects must be taken into account during the writing of the text.
Response: The following paragraph has been inserted to emphasize this point in the introduction apart: “The innovative character of the present work is based on the study carried out for the first time to determine the behavior of the four main phenolics of the alperujo over four points of the season for two consecutive years, as well as to promote the obtaining of a rich source of these phenols for industrial uses”
Below I indicate some points of improvement / suggestions.
- Line 81: Olive oil mill (OOM)
Response: The acronym of OOM has been modified in all the documents.
- The authors should detail in the introduction, which are the studies already published in the literature on phenolic composition in the matrix in question (olive pomace)
Response: A phrase has been introduced to express which phenols have been identified and quantified as major phenols by stating their concentration: “The main phenolic compounds identified and quantified in the alperujo and in the solid obtained in the three-phase extraction system are HT, Ty and their precursors, mainly oleuropein, which is transformed very rapidly as early as the malaxation. Therefore, HT and Ty are considered to be the main simple phenols in these by-products in quantities of about 300 and 450 ppm respectively [8]”.
- Authors are encouraged to detail the methodology, explaining how the quantification of phenolic compounds was performed. Are they expressed in calibration curves, expressed in mg tyrosol equivalent as recommended by the IOC?
Response: These aspects have been clarified in 3.6 section of materials and methods.
- The behavior observed (season 1 and 2) in the concentration of the phenolic compounds in question should be further discussed.
Response: To improve this aspect the following paragraph has been insert: “In general, the differences in phenolic content for the same cultivar depend on agronomic variations and the incidence of light and water in each year. In drier years there is a stress that causes the phenolic content to increase, as well as if the plant has been attacked by phytopathogens or even insects. But in this case, the difference found with DHPG or Ty is more important, which could be due to the similarity of DHPG with phytohormones and to the fact that Ty precursors are present in smaller quantities in olives and can undergo these changes more significantly [9,10,12,24]”.
- Figure 1: mg T/Kg fresh alperujo? The graphs are confused. It is difficult to understand the results. Please, think in the way to clarify better the results obtained.
Response: There has been an error in the units which has been corrected in figures 2, 3, 4 and 5. The units in which the phenols are expressed are mg/kg fresh pomace except for the concentration of phenols in the liquid fractions obtained as phenol sources where they are expressed in mg/L. All this has been clarified in section 3.6 of materials and methods.
- All graphs and tables deserve improvement. The results obtained may even be interesting, but both the methodologies and the results obtained are lacking in detail.
Response: The units have been clarified and all data discussed, and the discussion and conclusions have been improved.
- Sometimes mention “olive oil solid waste”, “pomace”, “alperujo”, please standardize the writing.
Response: The word “pomace” has been replaced for “alperujo” except for the use of pomace extractor. To clarify this point, the following sentence has been inserted in the introduction: “The main current use of alperujo is the extraction of pomace oil in pomace extractors, and once the oil has been obtained, the solid is used as biomass for energy production”. The expression "olive oil solid waste" is only put in the title to refer to "alperujo" which is how it is referred to throughout the document.
- 301: It is not good to start sentences with numbers.
Response: It has been corrected.
- Figure 4: The results are expressed in mg/L or mg/kg?
Response: It has been corrected previously.
- How were the quantifications of phenolic compounds performed?
Response: It has been clarified in the apart 3.6 of materials and methods.
- References must be formatted according to the journal's rules
Response: When we changed to the word format of the magazine, the italics and bold letters were lost, this has been corrected.
- All abbreviations must be cited in the text
Response: All the Abbreviations are now cited in the text.
- All figures and tables must be cited in the text
Response: All the figures and tables are now cited in the text.
- It is missed the critical point of view in conclusion section.
Response: The critical point of view has been improved in the conclusion inserting the following paragraph: “Further studies to increase the content of these phenolic compounds of high func-tional interest will be necessary. Thus, in order to improve the processes that aim to im-prove the quality of olive oil, they must also improve the quality of the by-products, such as olive pomace, by increasing and facilitating the extraction of their bioactive com-pounds. It is very important to emphasise that this type of study is only the beginning, since in order to make the extraction and use of these components possible, the synergy of different technologies must be found in order to achieve the integral use of the by-product [7,14,31].”
- The cited references are relevant, but I miss a critical look at the results indicated by the literature.
Response: New references have been inserted to improve this point.
Round 2
Reviewer 2 Report
I consider that the manuscript can be accepted in its current version, since the authors substantially improved its content based on the reviewers' recommendations.
Author Response
We thank the reviewer for his comments which have helped to improve the quality of the work. Some grammatical errors in English have also been corrected.
Reviewer 3 Report
Dear authors,
Thank you for you review. However, some more aspects must be clarified for the manuscript to be accepted:
Mention the concentration range of the calibration curve.
Detail the suppliers of each standard used.
Mention how the standard solutions were prepared (diluted in - water, methanol, acetonitrile...?)
R2 ≥ 0.99??
It would be important to mention the possible ways of "transforming" oleuropein compound into hydroxytyrosol and tyrosol, hypothetically indicating what could have happened in the case of your study, based on the literature reports. Additionally, perform a critical analysis of the results obtained in different seasons.
Author Response
Mention the concentration range of the calibration curve.
Response: It has been insert in the text: “The quantification was performed by means of a five-point regression curve in triplicate of individual stock solutions of each phenolic in the range of 10 to 1000 mg/L”.
Detail the suppliers of each standard used.
Response: It was previously showed in the apart 3.2 of Material and Methods.
Mention how the standard solutions were prepared (diluted in - water, methanol, acetonitrile...?)
Response: The following mention has been insert in the text: “A standard solution was prepared for each phenol at a concentration of 1000 mg/L by dilution of the standard in distilled water”.
R2 ≥ 0.99??
Response: It has been removed.
It would be important to mention the possible ways of "transforming" oleuropein compound into hydroxytyrosol and tyrosol, hypothetically indicating what could have happened in the case of your study, based on the literature reports. Additionally, perform a critical analysis of the results obtained in different seasons.
As these are samples of alperujo harvested from the same mills in two consecutive years, the differences in the concentration of the phenols studied must be due to agronomic differences and not to differences in the variety or maturity of the fruit. This point is already explained in the text. The agronomic factors that change for the same area in two different seasons are temperature, irrigation and light incidence, and it is precisely these factors that have been identified in the literature as capable of increasing or decreasing the phenolic content. In other words, under stress conditions (cold temperatures, water or sunlight) the tree synthesises more phenols for its defence. Another factor is that there has been a greater presence of phytopathogens in one of the seasons, which would also justify a change in the amount of phenols. When the phenol concentration increases or decreases, phenols are always the precursors of the four phenols studied, but the reactions for the formation of these phenols are always the same. Therefore, the content of HT, Glu-HT, Ty and DHPG will not depend on the type of transformation of their precursors but on their initial concentration. The initial concentration of the main precursors (oleuropein, ligustroside, etc… ) depend on the ripeness, the variety, the infections of the tree and the agronomic conditions. Thus, the infections and the agronomics conditions must be, in this case, the main reason of the difference between seasons. This point is explaining in the text.